# Factors related to the utilization of digital adherence technologies in tuberculosis care: A qualitative study among adults in DS-TB treatment, health care providers and other key actors in Tanzania

**Bianca Gonçalves Tasca**[1]*, **Andrew Mganga**[2], **Chung Lam Leung**[1], **Lucas Shilugu**[2], **Christopher Pell**[1], **Baraka Onjare**[2], **Nicholaus Luvanda**[2], **Liberate Mleoh**[3], **Liza de Groot**[1], **Kristian van Kalmthout**[1], **Katherine Fielding**[4], **Degu Jerene**[1]

1 KNCV Tuberculosis Foundation, The Hague, Netherlands, 2 KNCV Tuberculosis Foundation, Dar es Salaam, Tanzania, 3 National Tuberculosis and Leprosy Program, Ministry of Health, Dodoma, Tanzania, 4 TB Centre and Department of Infectious Disease Epidemiology, Faculty of Epidemiology and Population Health, London School of Hygiene & Tropical Medicine, London, United Kingdom

* bianca.tasca@kncvtbc.org, biatasca@gmail.com

**Data Availability Statement:** All participants' descriptive data and selected quotes are in the

## Abstract

Numerous challenges, such as lengthy treatment course, side effects, and distance to healthcare facilities contribute to suboptimal Tuberculosis (TB) treatment completion. Digital adherence technologies (DATs), such as smart pillboxes and medication labels, could be an alternative to facilitate TB treatment continuation. In-depth interviews with people undergoing treatment for drug susceptible TB, health care providers (HCPs) and other key actors were conducted to evaluate their experiences with DATs in ten health facilities across four different regions in Tanzania. This study is part of a multi country cluster randomized trials conducted under the ASCENT consortium. Interviews were conducted with a total of 41 individuals, 19 people with TB and 22 HCPs and key actors. One of the main findings indicates that participants found that the daily reminders provided by the DATs, particularly the alarm from the smart pillboxes, helped in supporting treatment continuation and establishing a routine around medicine intake. DATs use was linked with reducing the financial burden of treatment, improving people with TB-HCPs relationship, and decreasing workload for HCPs. Although DATs were generally perceived as reliable, occasional technical malfunctions, such as battery depletion in smart pillboxes, were reported. The requirement of having access to a cellphone and a power source emerged as specific barriers for medication label users. This study highlights the initial willingness and sustained enthusiasm for using DATs among respondents. DATs were perceived as useful tools, aiding individuals with treatment continuation through daily reminders and fostering stronger connections with HCPs. Nevertheless, issues such as poor network connectivity and the need for access to a working cellphone posed difficulties in usage. Findings from this study suggest the potential for improvements in the technologies and indicate that a thorough assessment of people's life

manuscript. Primary qualitative data such as transcripts could potentially be a breach of the confidentiality that the participants were promised upon request for participation. With that, authors are unavailable to provide interview transcripts.

**Funding:** This work was supported by UNITAID (Grant Agreement Number: 2019-33-ASCENT to KvK); Funder Website: https://eur01.safelinks.protection.outlook.com/?url=https%3A%2F%2Funitaid.org%2F%23en&data=05%7C02%7C%7C7b9927750170457c371408dca6884470%7Cf28ee6e5b8a648c589d0aaeef9ebd92e%7C0%7C0%7C638568352639117806%7CUnknown%7CTWFpbGZsb3d8eyJWIjoiMC4wLjAwMDAiLCJQIjoiV2luMzIiLCJBTiI6Ik1haWwiLCJXVCI6Mn0%3D%7C0%7C%7C%7C&sdata=%2Fdb%2BnkETSQ2XtcjwE4fBt3agizTnYeXldxl8fKzKy50%3D&reserved=0 The funders had no role in the study design, data collection and analysis, decision to publish, or preparation of the manuscript.

**Competing interests:** The authors have declared that no competing interests exist.

conditions and needs prior to treatment initiations is important to determine the suitability of providing a DAT.

## Introduction

Tuberculosis (TB) continues to be a major global public health challenge, affecting an estimated 10.6 million individuals worldwide and resulting in 1.3 million deaths in 2022 [1], despite being preventable and curable. Identifying and promptly treating individuals affected by TB and ensuring they have optimal conditions to complete the treatment has the potential to save millions of lives and eliminate the transmission of TB [2].

With around half the people with TB in the 30 high-burden countries successfully completing their treatment, there are concerning gaps in the TB care cascade [3]. Lengthy treatment duration and potential side effects are known to contribute to treatment discontinuation [4]. Other factors, such as forgetting to take medication, lack of knowledge about the importance of completing the treatment, drug stockouts, distance to the healthcare facility and stigma, also negatively impact treatment continuation [5, 6].

Directly observed therapy (DOT) has been the standard recommendation to enhance completion in TB treatment [7]. However, implementing strict DOT has posed challenges over the years. Frequent visits to clinics increased the workload for healthcare providers (HCPs) and created barriers for individuals on treatment due to extra costs associated with frequent travel [8]. In light of these challenges, digital adherence technologies (DATs) have emerged as possible alternatives to enhance TB treatment completion and improve outcomes. Recognizing the potential of digital solutions, the World Health Organization (WHO) issued recommendations regarding the implementation of DATs as part of integrated patient care [9].

DATs, such as cellphone-based reminders and smart pillboxes, have demonstrated the potential to enhance treatment initiation and completion in TB care [4]. A prior study conducted in Tanzania, involving people living with TB and HCPs utilizing medication labels (99 DOTs) for treatment support, revealed that HCPs believed the technology aided them in delivering better treatment, while people living with TB concurred that it facilitated their treatment continuation [10]. However, more in-depth information regarding user-related factors and the usage of DATs in TB treatment is needed to strengthen the evidence base.

For DATs to be effective, they must accommodate the needs and preferences of both people with TB and HCPs. In order to understand the factors that influence experiences with using and implementing medication labels, smart pillboxes and the EverWell adherence platform, we conducted in depth interviews with people living with drug susceptible TB (DS-TB), HCPs and other key actors in Tanzania.

## Methods

### Study setting

This qualitative study is part of a multi-country, cluster-randomized trial, the Adherence Support Coalition to End TB (ASCENT) consortium [11]. The trial was conducted in Ethiopia, Tanzania, The Philippines, South Africa, and Ukraine. Participants were randomized to standard of care versus intervention arms, composed of smart pillboxes or medication labels and differentiated care. Preliminary results in Tanzania show no difference between arms on the risk of poor end of treatment outcome, defined as either treatment failure, death, loss to follow-up, or a switch to a multi-drug resistant regimen [11].

The ASCENT cluster randomized trials were conducted from 2021 until 2023. Two separate randomization processes occurred prior to the commencement of the study. Firstly, health facilities were randomized to either the intervention or the standard of care arm. A second randomization process determined the type of DAT provided at each intervention arm facility. People with TB enrolled in the intervention arm were offered either a smart pillbox or medication labels, depending on the health facility they received care from. The DATs were offered for the entire duration of the participants' treatment period. If a person didn't have access to a cellphone in a health facility that provided medication labels, then a smart pillbox would be offered as an alternative solution. HCPs used the EverWell adherence platform to monitor DATs engagement, a proxy for adherence to treatment regimens.

Smart pillboxes are devices for storing TB medications. They provide users with audio-visual reminders and maintain a record of each instance the box is opened, transmitting this dosing history to the adherence platform. In the ASCENT intervention, the smart pillboxes were refilled every month by the HCPs during participants monthly visits to the health unit. Conversely, medication labels consist of blisters packs with printed codes. These codes are used by people with TB to report their medication intake via toll-free SMS texts. If a person living with TB fails to open the box or sends an SMS text—proxies of medication intake—an automatic message is dispatched from the adherence platform to their cellphone (or cellphone they have access to) as a reminder to take their prescribed medicine.

For HCPs, the adherence platform offered daily adherence data accessible through a web-based dashboard or smartphone app. In cases where people with TB were observed to have missed their medication, HCPs were instructed to initiate a differentiated care approach. This approach guidelines included sending cellphone message reminders for participants that missed one dose and a telephone call or home visits for participants missing multiple consecutive doses.

Prior to the start of the intervention, healthcare workers received training on the proper use of DATs and the adherence platform. Training was provided once for two days at the beginning of the project for all HCPs. ASCENT staff also provided monthly technical support on demand throughout the trial implementation period, including quarterly supportive supervision.

In Tanzania, DATs were implemented in 36 health facilities [11] as part of the ASCENT intervention. With an estimated population of 58 million in 2019 [12], the country is among the three high-TB burden countries that reached or surpassed the initial milestones of the End TB Strategy for reducing both TB incidence and deaths by 2019. Tanzania, however, is still categorized as a high-burden country for TB and HIV-associated [13]. Studies conducted in the country have shown suboptimal treatment completion rates, with only 79% of people with TB achieving 95% adherence levels in the Kilimanjaro [9].

TB treatment in Tanzania is administered through the primary healthcare network under the guidance of the National TB Programme (NTP). Tanzania's standard of care for DS-TB consists of a six-month course of first-line TB medications. The NTP recommends community or home-based DOT over facility-based DOT or unsupervised treatment when treatment supervision is deemed necessary [14]. During the ASCENT intervention, DOT was interrupted or provided intermittently due to the COVID-19 pandemic restrictions.

The treatment approach in Tanzania emphasizes patient-centered care, with a predominance of self-administered and home-based treatment modalities. The provision of free TB care in Tanzania aims to ensure universal access to treatment and mitigate the socioeconomic burden associated with the [12].

## Study population

In this qualitative study, respondents encompassed people with TB using DATs for their DS-TB treatment at ASCENT facilities, HCPs, and other keyactors, from the four study regions in Tanzania: Arusha, Geita, Manyara and Mwanza. Interviews were conducted in 10 randomly selected facilities, ensuring diverse representation based on the type of DAT used. Recruitment of people with TB occurred through a convenience sampling strategy between April 1st, 2022, and July 30th, 2022. HCPs and other key actors were recruited from January 22nd, 2023, to March 31st of the same year. Participants were invited to join the study via phone calls, with people with TB contacted by HCPs, and HCPs and other key actors contacted by ASCENT staff. All interviews were conducted by ASCENT research staff.

Eligibility criteria for people with TB included being at least 18 years old, receiving treatment for DS-TB, and using a DAT for a minimum of one month at the time of the interview. This time frame was chosen because it was considered that a minimum of one month was necessary for participants to provide meaningful input based on their experience.

Exclusion criteria encompassed individuals diagnosed with multi-drug-resistant TB (MDR-TB). HCPs were invited to participate in the interviews if they were over 18 years old and held positions as clinical officers, nurses, or community health workers at one of the 36 health facilities involved in the intervention arm. Additionally, professionals from the Ministry of Health/National TB program and ASCENT research staff were also invited to partake in the interviews.

An initial sample size of 20 people living with TB and 20 HCPs and other key actors was established. Interviews were conducted until research interviewers perceived data saturation.

## Data collection

In-depth interviews with people with TB took place during their routine visits to health facilities, where participants were informed on the purpose of the interviews, introduced to the researchers, and provided with a private and quiet space for the interviews. Key actors and HCPs interviews were prearranged and conducted in private office spaces to maintain confidentiality. A third person was present in the interview rooms only when participants required assistance, such as elderly participants accompanied by a caregiver.

Each participant was interviewed once and interviews followed a pre-tested interview guide, which consisted of open questions drawn from key themes and used in a flexible manner to capture each participant's unique experiences.

Among individuals living with TB, interviews explored topics such as, participants' social relations, barriers and facilitators to treatment completion, technology literacy, and their understanding and appraisal of the DATs. Interviews were conducted from February to September 2022. Within the group of key actors and HCPs, interviews explored their experiences in delivering TB treatment through DATs, using the adherence platform, implementing differentiated responses, and their perspectives on the feasibility of integrating DATs within the differentiated model of care.

Interviews were conducted by three male researchers in Kiswahili and audio recorded using portable sound recorders. Interviewers, AM, LS and NL were experienced in conducting qualitative interviews with TB/HIV clients. They had received post-graduate training in research methods and were oriented on the use of the interview guide.

## Data processing and analysis

Interviews recordings were transcribed verbatim and subsequently translated into English by AM and LS. The analysis process was undertaken by three additional researchers, BT, LG and

AL, with regular discussions with AM and DJ. Initially, all interview transcripts were thoroughly read by both researchers to gain familiarity with the content. Subsequently, relevant codes were developed, taking into consideration the study's objectives. Once patterns in the data were identified, themes were defined, and thematic analysis sessions were conducted to achieve a final consensus on the relevant themes. The selection of themes was based on their thematic relevance to the research aim [15]. The data were analyzed using NVivo 12 qualitative analysis software and Microsoft Excel.

For this study, a practical thematic analysis approach was adopted, employing a hybrid approach [16]. This approach entailed the utilization of both inductive and deductive methods, along with semantic and latent thematic analysis techniques, to develop the codes and derive meaningful insights from the data. The COREQ checklist was utilized for reporting the study findings.

## Ethical approval

Ethical approval to the study was obtained from the World Health Organization (WHO) Ethics Review Committee, the London School of Hygiene & Tropical Medicine (LSHTM), and Tanzania's National Institute for Medical Research (NIMR) as a sub study under the overall ASCENT protocol. Prior to participation, written informed consent was obtained from all participants, and the sessions were audio recorded for transcription purposes. During the informed consent process, participants were informed they could withdraw from participating at any moment and also request their interview transcripts for comments or correction.

## Inclusivity in global research

Additional information regarding the ethical, cultural, and scientific considerations specific to inclusivity in global research is included in the S1 Checklist.

## Results

Of the 41 respondents interviewed, 19 were people with TB, and 22 were HCPs and other key actors. Among people with TB, five were using medication labels and 14 were using the smart pillbox at the time of the interview. One of the participants using the smart pillbox had transitioned from the medication label during the trial due to cellphone damage. Among the people with TB who were interviewed, 11 (58%) were males and eight (42%) were between 30–39 years of age. In three instances people with TB were interviewed alongside their caregivers. This approach was adopted to address language barriers and/or assist elderly participants.

Key actors and HCPs who took part in the research comprised community health workers, nurses, district TB and Leprosy coordinators and ASCENT research staff. Key actors and HCPs participating in the study had a diverse range of years of experience in their respective roles, going from 1 to 14 years. Table 1 shows the characteristics of the interviewees.

In the context of the ASCENT intervention, nurses reported handling tasks such as enrolling people with TB in the adherence platform, and initiating them on treatment, whereas community health workers focused on treatment follow-up and home visits. Other key actors, including the Regional TB and Leprosy Coordinator, Regional TB and HIV Officer, and District TB and Leprosy Coordinator, described their main activities as ensuring DATs availability, organizing technical support, and providing oversight in ASCENT facilities. Meanwhile, ASCENT research staff managed routine activities like supportive supervision, monitoring treatment continuation via the EverWell adherence platform, communication with service providers, data collection for project studies, and offered technical support for the EverWell and DATs utilization.

**Table 1. People with TB, HCPs and key actors ' demographic characteristics.**

| People with TB | N (%) | | |
|---|---|---|---|
| **DAT type used at the moment of the interview** | | | |
| Smart Pillbox | 14 (74%) | | |
| Medication Label | 5 (26%) | | |
| **Sex** | | | |
| Male | 11 (58%) | | |
| Female | 8 (42%) | | |
| **Age range (years)** | | | |
| 18–29 | 2 (11%) | | |
| 30–39 | 8 (42%) | | |
| 40–49 | 5 (26%) | | |
| 50–59 | 0 (0) | | |
| $\geq$60 | 4 (21%) | | |
| **Total** | **19** | | |
| **Key actors and HCPs** | **N (%)** | | |
| Community health Worker | 4 (18%) | | |
| Nurse | 8 (36%) | | |
| District TB and Leprosy coordinator | 3 (14%) | | |
| ASCENT Research Staff | 1 (5%) | | |
| Regional TB and Leprosy coordinator | 4 (18%) | | |
| Regional TB and HIV officer | 2 (9%) | | |
| **Total** | **22 (100%)** | | |
| | Min. | Mean | Max. |
| HCPs and other key actors years of experience | 1 | 5 | 14 |

## Overall engagement with DATs

People with TB and HCPs demonstrated an initial inclination to engage with DATs. Amongst respondents, it was widely reported that the technologies could make one's life easier and participants were willing to try the DATs. In the HCPs interviews, respondents described feeling a renewed motivation to work once they heard about the implementation of the DATs. The interviewed HCPs also reported a high rate of technology uptake among people with TB. People with TB described a willingness to adhere to treatment and seemed to understand the importance of concluding the full treatment course.

> *"When patients come for starting TB treatment, they ask I hear there is this smart pillbox, I would love to have it for helping me with my medication." (*Nurse, #3516)

The ASCENT research team was responsible for training HCPs who subsequently instructed people with TB on using the DATs. HCPs indicated that these orientation sessions varied from fifteen minutes to two hours, depending on individual needs. Nurses assumed the responsibility of conducting these instruction sessions.

People with TB said that they were satisfied with the instructions received. Many participants recounted how adapting and incorporating the DATs into their daily routines was easy. Both technologies were referred to as user-friendly, demanding minimal effort. Frequently, they had acquaintances with prior experience of using DATs, which not only helped them better comprehend the technology but also provided support in effectively using DATs.

*"It was easy for me to understand because one of my friends had already used the technology before and he had explained the details to me; it helped me to be more accepting of using the technology. So, I understood the health provider very easily."* (Person with TB using medication labels, Male, Age: 30–39 years, #1908)

*"There is no challenge at all. It is easy to open, take the pills and close it. [There are] even no difficulties in storing it."* (Person with TB using a Smart Pillbox, Female, Age: 40–49 years, #2374)

## The perceived utility of DATs

### Daily reminders and treatment routine

People with TB emphasized that the alarm notifications and messages helped with treatment continuation, averting missed doses. The daily reminders helped them create a structured routine for medicine intake. People with TB said that just by seeing the smart pillbox they would already remember to take the medicine. Similar feelings were shared when respondents considered the viewpoint of their family members. They recounted how their relatives got involved in the treatment process by reminding them to take their medicine whenever they saw the box or heard its alarm.

*"Even myself by observing the box (smart pillbox) around, even if it is not yet time for taking the medication, I will remember that this box reminds me to take medication when it reaches a particular time"* (Person with TB using a Smart Pillbox, Female, Age: 40–49 years, #1535)

Nonetheless, some people with TB expressed a desire to have the flexibility to change the alarm volume for the smart pillboxes because there were instances when they could not hear the alarm sounding. Participants using medication labels did not benefit from alarms and healthcare workers mentioned that people using this technology would occasionally forget to send SMS notifications despite having taken their medication.

*"Mainly there are challenges with the 99 DOTS [medication labels], some [people with TB] most of the time they forget to send SMS, but they may have taken the medicines already."* (District TB and Leprosy coordinator, #3819)

HCPs and other key actors also pointed out that by engaging with a DAT, people with TB would become aware of the importance of staying on treatment and creating a routine around it. HCPs identified the adherence platform as another attribute of the DATs that helped people with TB in continuing the treatment because it helped prompt an action from their side and prevent people living with TB being lost to follow up.

*"The other benefit is that patients were not missing doses or dropping [out] from treatment, because we could easily notice that within a day or two and trace them to get the patient back [on their treatment]."* (Community Health Worker, #3314)

*"TB treatment takes six months. If taking medicine for a week tends to be a challenge, six months is a marathon. But through smart pillboxes, which remind patients when to take medication, the patients have been able to take their medicines daily and complete their treatment on time. And, in some cases, it does not only remind them to take their medication, but also in other activities, like waking up kids for schools, time to wake up for farming and milking the cattle."* (ASCENT project staff, #3011)

### Smart pillboxes as medicine storage devices

Participants from all groups highlighted that the smart pillboxes are a useful storage device for keeping medicine safe and away from dust and dirt. Nonetheless, some people with TB highlighted that the standardized features of the smart pillboxes did not consistently match their individual requirements. Some criticized the box for its compact size.

*"It [smart pillbox] enables me to store my medicine. (. . .) I could have been storing medicine using papers/magazine which is not very safe as the medicine could get dirty at times."* (Person with TB using a smart pillbox, Female, Age: ≥60 years, #2495)

### Cost-saving and reduced travels with DATs

Another perceived benefit arising from DAT utilization was the potential for cost savings. When individuals were using a DAT, they were only required to visit the health facility once a month to refill their medication, thereby reducing the necessity of additional trips to ensure medication intake. This also allowed people with TB to maintain their normal routine and provided them with additional time to rest, if needed. Key actors highlighted the importance of such an aspect for people living in remote areas and said this was an indication of how DATs could adapt well to different contexts.

*"It reduces the hustle of constantly coming to the health facility just for the [DOT provider] to see if you have really taken the medicine and saves travel costs to the health facility."* (Person with TB using Medication label, Male, Age: ≥60 years, #1498)

*"Among the benefits it has reduced financial costs for patients to visit facilities regularly particularly those residing at far places"* (Nurse, #4122)

### Managing care for the many

According to HCPs and key actors, the utilization of the adherence platform made the process of supporting treatment easier. Most HCPs reported being able to assess in a timely manner many people's treatment continuity patterns. This alleviated their workload because they did not need to rely on previously time-consuming strategies, such as counting empty blisters, to assure users were taking medication as expected. They felt as if the workload had diminished because they could perform tasks faster. Work also became more efficient because they could focus their efforts on people that showed suboptimal treatment continuation on the platform and prompt the initiation of a differentiated response based on the adherence platform information.

*"This new [adherence platform] has simplified our work. For instance, when you enter the office in the morning, you look on the tablet to monitor adherence of the patients on treatment, we follow up on patients with bad adherence who can say that they experience network challenges or maybe they forgot on that day, or s/he has been hospitalized on that day, so ASCENT has been so useful on monitoring patients for us." (Nurse, #4122)*

### Cellphone-related challenges and the use of medication labels

The requirement of access to a cellphone prevented the widespread provision of the medication labels to people living with TB in Tanzania. It was common for people to share cellphones, lose the device during treatment, or change numbers. These factors, together with the need for

access to a power source to charge the cellphone, made it challenging for people with TB to effectively engage with the medication labels.

*"The challenge is with charging the phone; some days it happens the phone has no power and has to be charged. As a result, you take the pills without sending the massage." (Person with TB using medication label, Male, Age: 40–49 years, #1498)*

*"The challenges are with {medication} labels because of using cellphone, you may find a client has lost their cellphone, and when they change for another SIM card, they won't tell you until you start follow up." (Nurse, #3516)*

### Technical issues and DATs

Overall both technologies were described as generally reliable but not without technical malfunctions. People with TB on both the smart pillboxes and medication labels said that sometimes they would receive a reminder message even after opening the box/sending a message, leading to confusion. Other technical challenges specifically related to the smart pillboxes were the fact that the alarm would ring at incorrect times and battery depletion, which required people to go to the facility for charging. HCPs noted that people with TB often became concerned when encountering errors in the DATs. In response, during the instruction sessions, they began explaining that technical malfunctions could happen, to alert and reassure users. The adherence platform also displayed errors from time to time, occasionally taking a long time to update medication intake information.

*"I got confused and I had to contact the nurse, the nurse said that I had to continue sending [confirmatory messages] to the number. I continued. . . most times I would get a message back [from the platform] other times I would get a message reminding me to take my medication in the evening." (Person with TB using Medication label, Male, Age: ≥60 years, #1806)*

*"Some days the adherence platform could not be accessible. This happens every now and then, so you may find red marks to all patient calendar as if patients have not taken their medicines while they did swallow because when we call them, they will say I did swallow the pills." (Nurse, #2506)*

Some people with TB faced intermittent network connection, which prevented them from being able to send or receive messages in a timely manner. HCPs described that such issues also affected users on smart pillboxes when the box could not synch with the platform until the person arrived in an area with network connection. This would make people with TB worry, and some mentioned going to the health facility to seek guidance or even sending several messages in a day to ensure their medication intake was accurately recorded.

*"The other challenge is poor network on some days, which led to the message failing to be delivered. It appears like you did not take the medicine, whereas in fact you did take it, but due to poor network the short text code could not be successfully sent." (Person with TB using Medication label, Male, Age: 40–49 years, #2014)*

### Social and interpersonal aspects of using DATs

### Enhanced people with TB-HCPs relationship

Among the perceived benefits of using DATs, some people with TB described feeling connected to HCPs facilitated by these technologies. They perceived HCPs as closer to them and

found an added value in terms of understanding that DATs facilitate HCPs tasks by automatically reporting their medicine intake history.

> *"The pill box has been helpful for quickly sending information to the nurse that I have taken medication on that day. Just after opening the box the nurse will get to know, I have taken the pills."* (Person with TB using Medication label, Male, Age: 30–39 years, #2301)

> *"Using this technology helps simplify communication with the clinic. I get to inform the nurse every day I take the pills, I see like I am together with the nurse."* (Person with TB using Medication label, Male, Age: ≥60 years, #2014)

Most HCPs echoed this impression, highlighting the positive impact on their relationship with people with TB. They phrased this in terms of building greater trust and collaboration with people with TB. Even though the information on the adherence platform can only be seen as a proxy for medication intake, they perceived that they could trust that people were properly taking the medication. Also, they described how people with TB could also have more trust in HCPs because they would be able to act quicker if they realized a person was not utilizing the technology.

> *"The intervention has made us work together with the patients unlike in the past when we just provided the medicines to the patient and let them go until they return for refill in a week or two. Even if they experienced challenges/side effects on taking their pills it was difficult to know. So, I see the patients feel as if they are valued and closely cared for. When we follow up and ask them about their progress, they say thanks for caring about my health, thanks for calling to ask about my condition. So, this intervention has given us a close and friendly relationship with our patients". (Nurse, #2203)*

## Social networks as influencers in DATs usage

People with TB described receiving encouragement to start using the DATs and continue their treatment. Some individuals mentioned familiarity with both the treatment and the DATs through friends and family previous experiences, which made it easier to understand and utilize the technology. People with TB themselves expressed a willingness to encourage other people to engage with the DATs.

> *"Interviewer: So, disclosing using a DAT was not a problem to you?*

> *Person with TB: No, the thing is once you start using that thing, you will convince other people to use it because it's sweet."* (Person with TB using Smart Pillbox, Male, Age: 40–49 years, #1663)

HCPs noted that some people with TB had concerns about being seen with the smart pillboxes. This feeling is particularly true among younger people who may feel 'shy' about carrying the devices. Additionally, HCPs have reported that people using the medication labels may have concerns that others could access their cellphone messages and discover that they are undergoing TB treatment.

> *"Some of the youth we registered to use pill boxes were worried on how to carry the pill box when they want to travel to somewhere else, unlike the elders who are just comfortable, the youth feel a bit shy or like how will people see me if I carry this pill box."* (Nurse, #2304)

### Cultural traditions and the reutilization of DATs

Cultural traditions were raised as a potential obstacle for the reuse of smart pillboxes. In Tanzania, there exists a custom of burying individuals along with their possessions, and this has raised apprehensions among most key actors about the feasibility of re-using the DATs, even though there seemed to be limited instances of such occurrence.

> *"There is also an understanding challenge: you may tell a patient that this smart pillbox given to you will need to go to another patient after you are done with treatment and the treatment supporters may be present hearing that, but if that patient dies. . . in some society's tradition, particularly some of our fellow Sukuma believe that when a patient dies, s/he should be buried with all of her/his treatment equipment, so these are among the challenges with DAT."* (ASCENT project staff, #3011)

## Discussion

Tuberculosis treatment is a lengthy process that can pose numerous challenges, such as adverse reactions, which can lead to treatment [17]. DATs were designed to provide an additional layer of support, aiding individuals to follow their treatment regimen. The objective of the present study was to evaluate the firsthand experiences of individuals living with TB, HCPs and other key actors within the Tanzanian healthcare system, regarding the utilization of DATs: the medication labels and the smart pillboxes alongside the adherence platform utilized by HCPs.

The findings indicate an initial willingness among participants to utilize DATs. HCPs indicated that utilizing DATs as part of their treatment was highly acceptable to people diagnosed with TB; HCPs themselves were motivated to integrate these technologies into their daily work routines. This attitude appeared to persist even after participants experienced the technologies, as evidenced by people with TB predisposition of recommending DATs to others. This indicates an acceptance of the studied DATs, which aligns with findings from other studies conducted in Uganda and Tanzania, where the 99DOTS technology (medication label) was found to be highly acceptable among people with TB, HCPs, and TB officers [10, 18]. Additionally, the smart pillboxes were also found to be associated with high degrees of satisfaction and acceptability among people with TB and healthcare workers in China [19].

During the interviews, engaging with DATs was generally seen as straightforward and requiring minimal effort. Training sessions seem to have had an important role in shaping this impression. Previous studies have also highlighted the need for training and follow-up advice sessions for appropriate engagement with the DATs, whereby sufficient orientation for HCPs, people with TB and their families was were a necessary condition for appropriate technology engagement [18, 20, 21].

One of the standout advantages offered by DATs was their ability to prompt medication reminders, a feature associated with enhanced adherence, as recounted by people with TB and HCPs. DATs accomplish this through functionalities such as the alarm and messaging systems, engaging not just people with TB but also involving caregivers and family members as seen in the interviews. This benefit was mostly perceived by users of the smart pillboxes. The adherence platform was also seen as an important determinant of treatment continuation because HCPs were able to target their actions to people with suboptimal treatment continuation. Similar findings were reported in a study conducted in India, where people with TB appreciated that smart pillbox reminders helped them maintain treatment consistency and avoid missing doses [21].

In the current study, participants reported a reduction in financial burden due to the DATs. This was linked to reduced journeys to the healthcare facilities and the ability to continue their regular daily activities. This is consistent with findings from prior studies in which HCPs highlighted that people with TB, especially from rural areas who usually must travel long distances to reach a health unit, benefited from less frequent travelling [20, 21]. This observation underscores the potential contribution of these technologies towards fulfilling a key objective of the End TB Strategy, which seeks to prevent individuals from incurring catastrophic costs related to TB [22].

The HCPs perception that DATs improved their work by enabling easy monitoring of individual's' medication intake and diminishing workload is another important perceived benefit of the DATs. In a previous study conducted in Tanzania, HCPs also reported a decrease in workload which enabled them to spend more quality time engaging with people with TB [10, 20].

Network issues, along with the need for a cellphone and access to electricity, posed a significant barrier for the uptake of the medication label. Such an issue was already reported as an important limitation of this technology and although in the present study it was not possible to assess differences between area of living and socioeconomic strata, studies indicate that people living in rural areas and of low socio-economic levels might be the most affected by these issues [18, 21].

In this study, DATs were seen as having the capacity to improve the relationship and collaboration between people living with TB and HCPs. People with TB have said to feel more connected to the HCPs and HCPs expressed feeling a greater sense of collaboration with patients. This was also reported in different studies where DATs appear to have contributed to the cultivation of mutual trust and care between individuals with TB and HCPs [10, 18, 23].

A small number of individuals with TB interviewed mentioned being reluctant to disclose their TB status or the usage of DATs. However, HCPs noted that younger participants expressed apprehension about using smart pillboxes due to concerns about potential stigmatization. This finding aligns with a study conducted in South Africa, where HCPs reported that people with TB struggled to keep their TB status concealed when using smart pillboxes. As a result, these individuals often left the devices behind when they needed to travel or go to work [21, 24].

Cultural traditions were viewed as a potential obstacle to the reuse of DATs. Key actors commonly believed that burial customs could be an obstacle to the sustainability of DATs implementation even though there was only one occurrence reported. Nevertheless, conducting further investigation and providing specific guidance on this matter could be a valuable initiative in this context.

## Strengths and limitations

The study is limited by the fact that participants were recruited through convenience sampling from health facilities, which mean that the perspectives of individuals that discontinued using DATs or were lost to follow up were not included. We can only partially grasp the impressions of the people with TB who experienced difficulties with their treatment regimen or with the DATs through the information that HCPs and key actors provided. This also prevented us from gaining further insight into the differentiated response because only individuals with suboptimal treatment continuation would receive such care. Moreover, only five participants using medication labels were interviewed, which led to an unequal distribution of participants between the different DATs. Two key strengths of the study are the diversity of roles held by the interviewed key actors, which ensured a wide range of opinions, and the inclusion of

participants from the four regions of the trial, enhancing the study's representativeness within the Tanzanian context.

## Conclusions

This study aimed to provide insights into the use of medication labels, smart pillboxes and the EverWell adherence platform, for TB treatment in Tanzania among people with TB, HCPs and other key actors. Interviews suggest that respondents showed an initial willingness to accept the technologies, which was sustained after experiencing the tools.

The study findings underscore that DATs were seen as useful by participants but that the technologies' suitability is dependent on a person's life conditions and preferences. Factors such as owning a cellphone, access to reliable network and electricity, age, and socioeconomic status influence the extent to which a person can benefit from these technologies.

To establish a more patient-centric approach to care, the decision regarding which DAT to use and whether to employ one should be thoroughly discussed and evaluated in consultation with the person initiating TB treatment. Further research, stratified by socioeconomic strata, could provide more insights into how well-suited these technologies are for benefiting those who are most in need.

## Supporting information

**S1 Checklist. Inclusivity in global research questionnaire.**
(DOCX)

**S2 Checklist. COREQ (COnsolidated criteria for REporting Qualitative research) check-list.**
(PDF)

## Acknowledgments

The authors would like to thank all study participants for their valuable contributions.

## Author Contributions

**Conceptualization:** Bianca Gonçalves Tasca, Liberate Mleoh, Kristian van Kalmthout, Katherine Fielding, Degu Jerene.

**Data curation:** Andrew Mganga, Chung Lam Leung, Nicholaus Luvanda, Liza de Groot.

**Formal analysis:** Bianca Gonçalves Tasca, Chung Lam Leung, Degu Jerene.

**Funding acquisition:** Kristian van Kalmthout, Degu Jerene.

**Investigation:** Andrew Mganga, Lucas Shilugu, Nicholaus Luvanda, Katherine Fielding, Degu Jerene.

**Resources:** Kristian van Kalmthout.

**Supervision:** Baraka Onjare, Kristian van Kalmthout, Katherine Fielding, Degu Jerene.

**Validation:** Nicholaus Luvanda, Degu Jerene.

**Writing – original draft:** Bianca Gonçalves Tasca.

**Writing – review & editing:** Andrew Mganga, Chung Lam Leung, Lucas Shilugu, Christopher Pell, Baraka Onjare, Nicholaus Luvanda, Liberate Mleoh, Liza de Groot, Kristian van Kalmthout, Katherine Fielding, Degu Jerene.

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
