## [Decision Letter · Decision Letter 0]

23 May 2024

PGPH-D-23-02462

Factors Related to the Utilization of Digital Adherence Technologies in Tuberculosis Care: A Qualitative Study Among Adults in DS-TB Treatment, Health Care Providers and Other Key Stakeholders in Tanzania

Dear Dr. Gonçalves Tasca,

Thank you for submitting your manuscript to PLOS Global Public Health. After careful consideration, we feel that it has merit but does not fully meet PLOS Global Public Health’s publication criteria as it currently stands. Therefore, we invite you to submit a revised version of the manuscript that addresses the points raised during the review process.

Please see the comments of one reviewer below. Please note that we have only been able to secure a single reviewer to assess your manuscript. We are issuing a decision on your manuscript at this point to prevent further delays in the evaluation of your manuscript. Please be aware that the editor who handles your revised manuscript might find it necessary to invite additional reviewers to assess this work once the revised manuscript is submitted. However, we will aim to proceed on the basis of this single review if possible. 

We look forward to receiving your revised manuscript.

Kind regards,

Hanna Landenmark

Staff Editor

Journal Requirements:

Additional Editor Comments (if provided):

Reviewers' comments:

Reviewer's Responses to Questions

**Comments to the Author**

1. Does this manuscript meet PLOS Global Public Health’s publication criteria? Is the manuscript technically sound, and do the data support the conclusions? The manuscript must describe methodologically and ethically rigorous research with conclusions that are appropriately drawn based on the data presented.

Reviewer #1: Yes

2. Has the statistical analysis been performed appropriately and rigorously?

Reviewer #1: N/A

3. Have the authors made all data underlying the findings in their manuscript fully available (please refer to the Data Availability Statement at the start of the manuscript PDF file)?

Reviewer #1: No

4. Is the manuscript presented in an intelligible fashion and written in standard English?

Reviewer #1: Yes

5. Review Comments to the Author

Reviewer #1: Thank you for the opportunity to review this important research that contributes to the lack of understanding of participant and health care provider experiences using digital adherence technologies. Overall, well written and adds to the literature. The following are minor suggestions by section.

Abstract:

- The term ‘stakeholders’ is a common term, but it is currently recommended to use alternative terms - https://www.cdc.gov/healthcommunication/Preferred_Terms.html

Introduction

- Clear and well written

- Would be helpful to report on main findings of the trial to provide context. Or data on DAT usage patterns.

Methods

- Can the standard of care be more clearly defined

- More information about the differential care approaches could be helpful.

- Justification for inclusion of use of DAT for only one month is needed. There are likely different perspectives just starting versus using tool for longer time, e.g., 4-6 months. It might be noted in the discussion but rational for time selected is missing. And state that stage/month of use for each participant.

- Recommend using the COREQ guidelines for reporting qualitative study and report having used it to guide your report of the work. Seems that it was followed but not noted.

- What was the rationale for not using purposive sampling to get equal numbers for each intervention type? 5 seems like a low number to get saturation for the one DAT.

- How were the smart pillboxes filled (e.g., by whom and how often)?

Results

- It might add to know the number of participants experiencing/reporting the themes to have a better sense if a theme was experienced by just one participant or the majority.

- Results clearly written and informative. However, I was looking for more information on the actual usage of the DATs and if there were trends of use that could be corroborated with the qualitative. Maybe another paper?

- Also, results seem lacking in patients and HCPs perception of those who stopped using the DATs or had low reporting.

- Not a lot of discussion on the platform – does is show anything more than reporting calendars?

Discussion

- Do authors consider the uneven or low number of patient participants for one DAT type a potential limitation? I suggest describing either way if considered limitation or if not.

6. PLOS authors have the option to publish the peer review history of their article (what does this mean?). If published, this will include your full peer review and any attached files.

**Do you want your identity to be public for this peer review?** For information about this choice, including consent withdrawal, please see our Privacy Policy.

Reviewer #1: **Yes: **Sarah Iribarren

---

## [Editor Report · Decision Letter 1]

8 Jul 2024

Factors Related to the Utilization of Digital Adherence Technologies in Tuberculosis Care: A Qualitative Study Among Adults in DS-TB Treatment, Health Care Providers and Other Key Actors in Tanzania

PGPH-D-23-02462R1

Dear Ms Gonçalves Tasca,

We are pleased to inform you that your manuscript 'Factors Related to the Utilization of Digital Adherence Technologies in Tuberculosis Care: A Qualitative Study Among Adults in DS-TB Treatment, Health Care Providers and Other Key Actors in Tanzania' has been provisionally accepted for publication in PLOS Global Public Health.

Best regards,

Sanghyuk S Shin

Academic Editor